# Heritability estimation of osteoarthritis in the pig-tailed macaque (*Macaca nemestrina*) with a look toward future data collection

Peter B. Chi[1], Andrea E. Duncan[2], Patricia A. Kramer[2] and Vladimir N. Minin[3]

[1] Department of Statistics, California Polytechnic State University, San Luis Obispo, CA, USA
[2] Department of Anthropology, University of Washington, Seattle, WA, USA
[3] Departments of Statistics and Biology, University of Washington, Seattle, WA, USA

## ABSTRACT

We examine heritability estimation of an ordinal trait for osteoarthritis, using a population of pig-tailed macaques from the Washington National Primate Research Center (WaNPRC). This estimation is non-trivial, as the data consist of ordinal measurements on 16 intervertebral spaces throughout each macaque's spinal cord, with many missing values. We examine the resulting heritability estimates from different model choices, and also perform a simulation study to compare the performance of heritability estimation with these different models under specific known parameter values. Under both the real data analysis and the simulation study, we find that heritability estimates from an assumption of normality of the trait differ greatly from those of ordered probit regression, which considers the ordinality of the trait. This finding indicates that some caution should be observed regarding model selection when estimating heritability of an ordinal quantity. Furthermore, we find evidence that our real data have little information for valid heritability estimation under ordered probit regression. We thus conclude with an exploration of sample size requirements for heritability estimation under this model. For an ordinal trait, an incorrect assumption of normality can lead to severely biased heritability estimation. Sample size requirements for heritability estimation of an ordinal trait under the threshold model depends on the pedigree structure, trait distribution and the degree of relatedness between each phenotyped individual. Our sample of 173 monkeys did not have enough information from which to estimate heritability, but estimable heritability can be obtained with as few as 180 related individuals under certain scenarios examined here.

Corresponding author
Vladimir N. Minin, vminin@uw.edu

## INTRODUCTION

Osteoarthritis is a condition that is characterized by the breakdown of cartilage in joints between bones, and can occur in any joint in the body. Those who suffer from osteoarthritis may experience pain and soreness in the affected area, and even a lack

of mobility, particularly in spinal osteoarthritis. Thus, spinal osteoarthritis is a serious worldwide public health concern, and a better understanding of this disease can lead to better treatment and care of patients who suffer from it (*Hadjipavlou et al., 1999*). This disease is characterized by several radiological features, including narrowing of the intervertebral disk space, bone spurs along the spinal cord (osteophytosis), and vertebral end-plate sclerosis (*Lawrence, 1969*). The conglomeration of these features is generally referred to as degenerative disk disease, or DDD (*Vernon-Roberts & Pirie, 1977*), although this term is also used to indicate the presence of a single one of these features (*Cohn et al., 1997*; *Lawrence, 1969*).

Specific aspects of DDD in humans have been well-characterized throughout the literature. For example, evidence for associations between DDD and various factors have been demonstrated, including age (*Frymoyer et al., 1984*; *Riihimaki et al., 1990*), body mass (*Riihimaki et al., 1990*), trauma (*Kerttula et al., 2000*), type and level of activity (*Caplan, Freedman & Connelly, 1966*; *Riihimaki et al., 1990*; *Videman, Nurminen & Troup, 1990*; *Videman & Battie, 1999*), and gender (*Jones, Pais & Omiya, 1988*; *Miller, Schmatz & Schultz, 1988*). Research in other mammals has corroborated the contribution of biomechanical stress to the development of DDD (*DeRousseau, 1985*; *Schultz, 1969*). Indeed, the bipedality and erect posture of humans has been assumed to be one of the primary causes of DDD in our species (*Bridges, 1994*; *Jurmain & Kilgore, 1995*; *Knusel, Goggel & Lucy, 1997*; *Schultz, 1969*; *Shore, 1935*).

Nevertheless, much is still unknown about the etiology of DDD. In particular, the extent to which genetics plays a role in DDD development has not yet been uncovered. Since there are safety concerns posed by radiography, the macaque monkey is often used as an animal model for humans in the study of bone diseases, due to its close genetic relatedness to humans (*Duncan, Colman & Kramer, 2011*; *Duncan, Colman & Kramer, 2012*). One may question its appropriateness for DDD as macaque monkeys are not bipedal, but this concern was addressed by *Kramer, Newell-Morris & Simkin (2002)*, who explored DDD specifically in the macaque species known as pig-tailed macaques (*Macaca nemestrina*), and concluded that they are indeed an appropriate animal model for DDD in humans.

Here, we use a population of captive pig-tailed macaques to explore the question of whether there is a genetic component to DDD. To this end, we examine whether DDD is heritable. Heritability is a statistically defined quantity that describes the degree to which a trait is determined by genetics. Heuristically, if a trait has high heritability, then individuals who are more related to each other would appear more similar to each other than average, with regard to this trait. While genotyping in humans is now cheap and ubiquitous, heritability estimation in primates is still often performed to determine whether a trait warrants genotyping in the animal model, with the goal of mapping genes that control the trait.

Methodologically, we are interested in a model for the trait that would allow for a transparent heritability estimation. A common assumption is that the trait follows a normal distribution. This is generally justified by the polygenic model, which postulates that complex traits are under control by several additive, independent loci, with similar

variances (*Fisher, 1918*). However, this assumption may be drastically violated in some real data problems. In particular, if the trait is ordinal with only a few categories, it is clear that the trait would not follow any distribution resembling normality. Likewise, normality may be violated even if there are many categories, but one category is severely over-represented. Such is the case with our trait distribution in the pig-tailed macaques.

Heritability estimation with discrete data was first demonstrated for the binary case by *Dempster & Lerner (1950)*, and extended to multiple ordered categories by *Gianola (1979)*, using transformations of a continuous trait. However, some intrinsic difficulties to these tasks quickly presented themselves. First, unlike continuous traits, the variance of a binary trait is closely tied to the mean or prevalence of the trait, and thus provides no useful information about the inherent biological variability of interest (*Burton, Bowden & Tobin, 2007*). Furthermore, the observed scale of an ordinal trait may not be additive (e.g., an observation of "4" may not be equal to twice the value of an observation of "2"), thus leading to biases in parameter estimates (*Gianola, 1982*; *Höschele, 1986*). Thus, some authors have completely abandoned transforming/thresholding continuous models and has attempted to estimate heritability of discrete traits directly under Poisson and negative binomial mixed models (*Foulley, Gianola & Im, 1987*). However, these models have their own drawbacks as well, including the issue of whether heritability is even well-defined in these contexts (*Matos et al., 1997*).

Here, we consider the threshold model for ordinal data (*Wright, 1934*). This model makes the assumption that the value of the ordinal trait is dictated by an unobserved latent variable with a normal distribution, referred to by *Wright (1934)* as the *liability*. For example, with a binary variable that has observed states of 0 and 1, the value for any given individual would be determined by whether that individual's value of the liability is above or below some threshold. We choose this model primarily because of its biological justifiability, through applying the polygenic model to the liability. That is, regardless of the distribution of the observed trait, if the phenotype is determined by many genetic loci, then it is plausible that an underlying normally distributed liability would exist.

A comprehensive Bayesian framework for heritability estimation under the threshold model was formulated by *Sorensen et al. (1995)*. Further work was done through the next 10 years, e.g., improving the MCMC convergence (*Cowles, 1996*), and extending the framework to a censored normal outcome variable (*Sorensen, Gianola & Korsgaard, 1998*). There was, however, no widely available and actively maintained open-source software to perform such analyses until the recent appearance of the `MCMCglmm` package in R (*Hadfield, 2010*). Thus, the time has come when biologists with ordinal data wishing to estimate heritability using the threshold model can begin to do so more easily than before.

Of course, ordinal data will not necessarily follow the threshold model, and although we obviously cannot "know" what the true distribution of any real data trait is, it is still useful to examine the statistical properties of heritability estimates under various scenarios and models, to reveal the consequences of misspecifying the distribution of the trait. That is, if the trait follows the threshold model with a liability trait that follows a normal distribution, but we incorrectly assume that the trait itself follows a normal distribution, how is the
estimation of heritability affected by this incorrect assumption? We examine this under a variety of scenarios and number of categories. We then examine heritability estimation on our actual dataset under these different models, which illuminates the concern of how much data are needed to obtain estimable heritability under this model. Thus, we conclude with an exploration of sample size requirements, which will be useful in guiding future studies.

## METHODS

### Heritability

When estimation of heritability is performed with pedigree data, the structure of these data allows for the identifiability of the quantities that define heritability. The kinship coefficient $\Phi$ and coefficient of identity $\kappa_2$, also commonly referred to as $\Delta_7$ (*Wright, 1922*; *Jacquard, 1966*), are two quantities well established by classical population genetics. Here, we have a matrix $\Phi$ whose components $\Phi_{ij}$ are defined as the probability at a given locus that two gene copies chosen at random from two individuals $i$ and $j$ are Identical-By-Descent (IBD), and $\kappa_2$ is also a matrix whose components $\kappa_{2_{ij}}$ are defined as the probability at a given locus that two individuals $i$ and $j$ share two gene copies IBD. Then, for any trait vector $\mathbf{Y}$ of measurements taken on individuals within the pedigree, the polygenic model (*Fisher, 1918*) posits that $\mathbf{Y}$ will have a multivariate normal distribution with covariance matrix

$$\Sigma = 2\sigma_A^2 \Phi + \sigma_D^2 \kappa_2 + \sigma_E^2 \mathbf{I}, \tag{1}$$

where $\sigma_A^2$ is the variance of the additive genetic effect, $\sigma_D^2$ is the variance of the dominant genetic effect, $\sigma_E^2$ is the variance of the environmental effect, and $\sigma_A^2 + \sigma_D^2 + \sigma_E^2 = \sigma_Y^2$ if there are no other effects to consider (such as household or maternal), and there is no interaction or correlation between effects (*Lange, 2002*). Heritability of the trait $\mathbf{Y}$ is then defined as the ratio of the additive genetic variance to the total variance of the trait: $h^2 = \sigma_A^2 / \sigma_Y^2$.

### Estimation under normality

The framework of the polygenic model then leads us to consider a multivariate normal model for the vector of trait values from the whole sample. Under this model, the partitioning of the covariance matrix in (1) allows for estimation of these variance components through maximum likelihood. Furthermore, in this framework it is easy to adjust for covariates, as we can state that $\mathbf{Y} \sim \mathbf{N}(\mathbf{X}\boldsymbol{\beta}, \Sigma)$, where $\mathbf{X}$ is an $(n \times p)$ matrix for $n$ individuals and $p$ covariates of interest (e.g., age, weight, gender), and then $\boldsymbol{\beta}$ is a $(p \times 1)$ column vector of mean components. The $\beta$s are nuisance parameters since our object of interest is still just the variance components, but incorporating them into the model allows for control over confounders. Thus, we can write the usual multivariate normal likelihood: $L(\boldsymbol{\beta}, \sigma_A^2, \sigma_D^2, \sigma_E^2) = (2\pi)^{-\frac{n}{2}} |\Sigma|^{-\frac{1}{2}} \exp(-0.5(\mathbf{Y} - \mathbf{X}\boldsymbol{\beta})^T \Sigma^{-1}(\mathbf{Y} - \mathbf{X}\boldsymbol{\beta}))$ where $\Sigma$ is explicitly partitioned into our variance components of interest; thus we have a tractable likelihood that we can attempt to maximize with respect to our parameters. Computational issues in solving for the roots of the likelihood equations for variance components estimation were addressed by *Lange, Westlake & Spence (1976)* and implemented in the MENDEL software

package (*Lange et al., 2001*). One could also proceed using restricted maximum likelihood for fitting linear mixed models, available in software packages such as `ASReml` (*Gilmour, Thompson & Cullis, 1995*); however, we do not consider this here.

## Threshold model: ordered probit regression

For ordinal data, a more realistic assumption than normality of the trait may be to assume that this trait is dictated by an underlying normally distributed latent variable. Then, an individual's value of the liability trait would determine which category that individual falls into for the observed trait. Formally, we consider the following model:

$$\mathbf{U} = \mathbf{X}\boldsymbol{\beta} + \mathbf{a} + \boldsymbol{\varepsilon}; \qquad P(Y_i = j) = P(t_{j-1} < U_i \le t_j), \tag{2}$$

where $\mathbf{U} = (\mathbf{U}_1, \ldots, \mathbf{U}_n)'$ is the vector of unobserved liabilities for each individual, and $\mathbf{a}$ is a random vector representing the breeding values for each individual, with $\mathbf{a}|\sigma_A^2 \sim N(\mathbf{0}, 2\boldsymbol{\Phi}\sigma_A^2)$. Then, with $\boldsymbol{\varepsilon} \sim N(\mathbf{0}, \sigma_E^2\mathbf{I})$, the latent variable vector $\mathbf{U}$ has the same covariance structure given by $\boldsymbol{\Sigma}$ in (1) above, assuming here that $\sigma_D^2 = 0$. Finally, $\mathbf{t} \equiv (\mathbf{t_0}, \ldots, \mathbf{t_C})$ are the true but unknown cutpoints on the distribution of the latent variable, which, along with the values of each $U_i$, determine the values of each $Y_i$, where $\mathbf{Y}$ is the observed categorical outcome vector. This forms the basis for Ordered Probit Regression (OPR).

Heritability estimation under this model could be performed through either Maximum Likelihood or Bayesian approaches. Since open-source implementation for Bayesian approaches to heritability estimation under this framework are readily available, we proceed in that manner. Namely, we use the R package `MCMCglmm` (*Hadfield, 2010*). Ideally, we would like to approximate the posterior distribution of $\sigma_A^2$ and $\sigma_E^2$ so that we can estimate heritability. Here, this is done along with concurrent estimation of $\mathbf{U}$, $\boldsymbol{\beta}$, and $\mathbf{t}$, given the data $\mathbf{Y}$ that we observed and the pedigree. We impose inverse gamma prior distributions on $\sigma_A^2$ and $\sigma_E^2$, with shape and scale parameters ($\alpha_A$, $\gamma_A$ and $\alpha_E$, $\gamma_E$, respectively) of 0.01. We note here that these distributions on the individual variance components impose a Beta(0.01, 0.01) prior distribution on $h^2$, which will be discussed later.

To facilitate the Gibbs sampling, data augmentation of the unobserved liability $\mathbf{U}$ is included as a latent variable, which we have already assumed to have a normal distribution, given $\boldsymbol{\beta}$ (*Tanner & Wong, 1987*; *Albert & Chib, 1993*). Then, the joint posterior distribution of the parameters and latent variables is given by:

$$p(\boldsymbol{\beta}, \mathbf{U}, \mathbf{t}, \sigma_A^2, \sigma_E^2|\mathbf{Y}) \propto p(\boldsymbol{\beta})p(\mathbf{t})p(\mathbf{U}|\boldsymbol{\beta}, \sigma_A^2, \sigma_E^2) \times p(\sigma_A^2|\alpha_A, \gamma_A)p(\sigma_E^2|\alpha_E, \gamma_E)p(\mathbf{Y}|\mathbf{U}, \mathbf{t}), \tag{3}$$

where most of these distributions have already been mentioned above, but the prior $p(\beta)$ follows a normal distribution with a variance of $10^{10}$ and appropriate dimensions for the number of fixed effects (e.g., age, weight), the prior $p(\mathbf{t})$ for the thresholds is flat and improper, and $p(\mathbf{Y}|\mathbf{U}, \mathbf{t})$ is simply a vector of indicator functions of whether each $Y_i$ falls into the category corresponding to the true value of $U_i$ and $\mathbf{t}$. To improve convergence, a Metropolis–Hastings-within-Gibbs strategy is implemented in `MCMCglmm`, where $\mathbf{U}$ and $\mathbf{t}$ are updated jointly using a Metropolis–Hastings step at each iteration, $\beta$ is sampled jointly

from the entire vector's full conditional distribution, and $\sigma_E^2$ and $\sigma_A^2$ are each sampled independently from their individual full conditional distributions (*Cowles, 1996*; *Hadfield, 2011*).

## Identifiability of variance components and heritability

In latent models with an ordinal response variable, individual variance components many not be identifiable (*Harville & Mee, 1984*; *Mizstal, Gianola & Foulley, 1989*; *Luo et al., 2001*; *Stock, Distl & Hoeschele, 2007*; *Ødegård et al., 2010*). A common solution to this problem is to fix one of the variance components to a known constant $c$ (e.g., $\sigma_E^2 = 1$). This solution is viable, because even when individual variance components are not identifiable, heritability—the main object of interest—may still be (*Stock, Distl & Hoeschele, 2007*; *Ødegård et al., 2010*). In our case, fixing $\sigma_E^2 = c$ allows us to re-parameterize our model in terms of heritability instead of variance components, yielding the following posterior distribution:

$$p(\boldsymbol{\beta}, \mathbf{U}, \mathbf{t}, h^2|\mathbf{Y}) \propto p(\mathbf{Y}|\mathbf{U}, \mathbf{t})p\left(\mathbf{U}|\boldsymbol{\beta}, \sigma_A^2 = \frac{h^2 c}{1 - h^2}, \sigma_E^2 = c\right)p(h^2), \qquad (4)$$

where $p(h^2)$ is density of the Beta distribution, as discussed in the previous section.

   Although the outlined approach to solving the identifiability problem is theoretically valid, in practice, fixing one of the variance components results in severe mixing problems of MCMC algorithms designed to approximate the posterior (4) (*Ødegård et al., 2010*). An alternative solution is to use MCMC to sample from the posterior of the unidentifiable model (3), but draw inferences based on only the posterior of heritability parameter, $h^2$. This latter approach can be viewed as MCMC with auxiliary variable augmentation of the state space, where $\sigma_E^2$ plays the role of an auxiliary variable. Using simulated data, we demonstrate that the auxiliary MCMC approach is superior in practice to the MCMC targeting the posterior (4), at least when using `MCMCglmm` package. Figure 1 shows traceplots of variance component(s) and heritability under both MCMC sampling schemes, using two different pedigree structures. For the first pedigree, fixing $\sigma_E^2 = 1$ results in such slow mixing that the Markov chain does not reach stationarity, while the auxiliary MCMC mixes very well, settling on the true value of heritability, which we set to 0.6 for both pedigrees. Using the second pedigree and fixing $\sigma_E^2 = 1$, we observe possible stationary behavior of the heritability traceplot, but still very slow mixing with 1000 MCMC iterations corresponding to an effective sample size of 15. The auxiliary MCMC mixes much faster with 1000 MCMC iterations corresponding to an effective sample size of 615. These two examples and results of our extensive simulation study, outlined below, demonstrate that the auxiliary MCMC, even though unconventional, appears to work well in practice.

## Data
### Simulations

We simulate several datasets under a variety of conditions. The simplest scenario is that of a three generation pedigree shown in Fig. 2, where trait data are simulated over 40

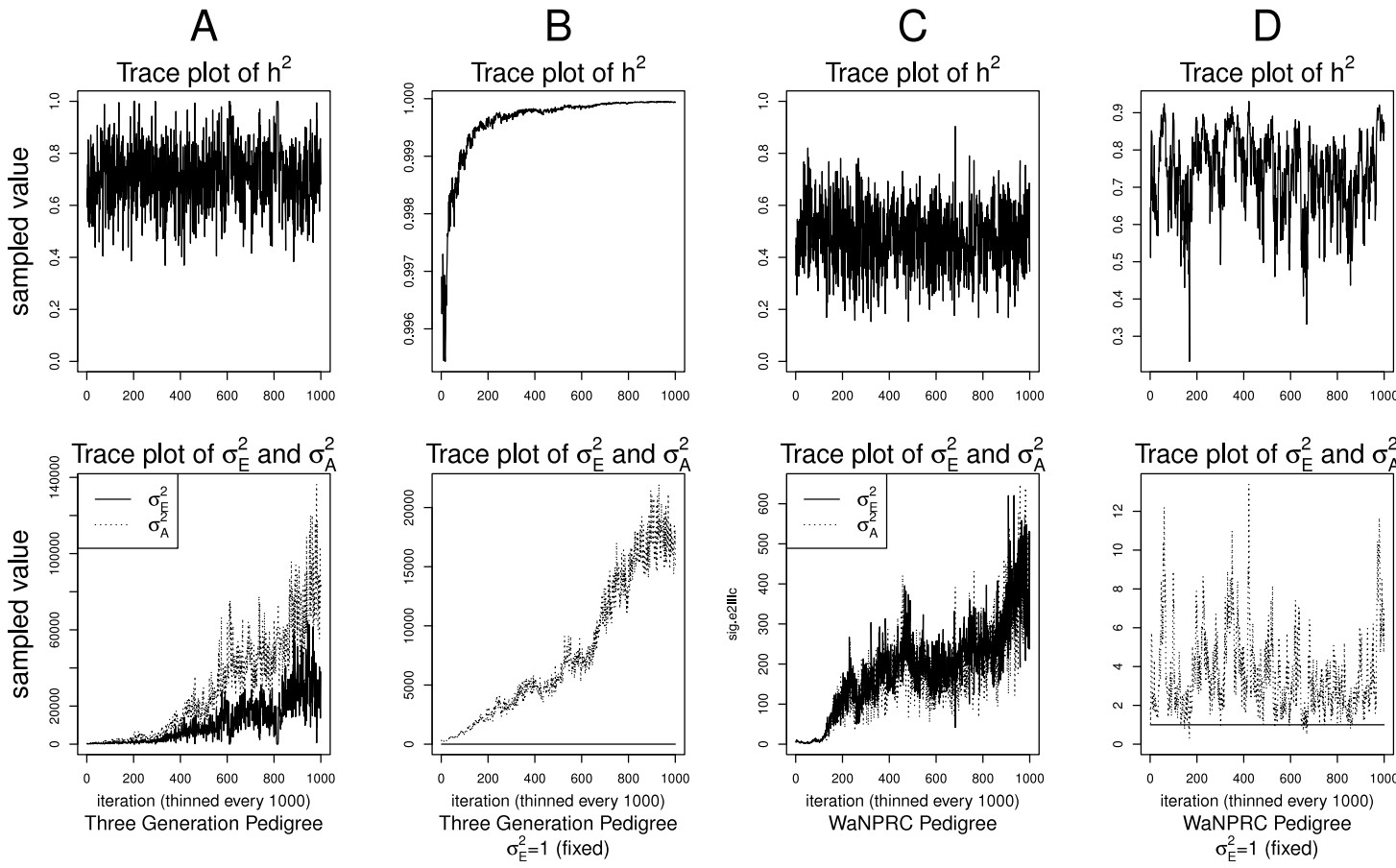

**Figure 1 Variance components and heritability traceplots.** Four scenarios are shown here, with traceplots of $h^2 = \sigma_A^2/(\sigma_A^2 + \sigma_E^2)$ on top, and traceplots of the individual variance components on the bottom. The first scenario (**column A**) is the three generation pedigree. While the MCMC samples of each individual variance component clearly do not show convergence (bottom), we observe that when we examine the corresponding values of $h^2$, this does appear to be stable (top). Conversely, when we fix $\sigma_E^2 = 1$, this does not appear to stabilize the MCMC samples of $\sigma_A^2$ here, and $h^2 \to 1$ as shown in the top and bottom panels of column B. With the WaNPRC pedigree (**C and D**), we again observe that without fixing $\sigma_E^2 = 1$, the MCMC samples of $h^2$ does indicate convergence despite the fact that those for $\sigma_A^2$ and $\sigma_E^2$ individually do not. On the other hand, when fixing $\sigma_E^2 = 1$, we observe that $\sigma_A^2$ does not "blow up" like it did in the three generation pedigree case, but mixing appears to be poorer with regard to the traceplot of $h^2$. Indeed, in these 1000 MCMC samples, our effective sample size is 15, compared to 615 when $\sigma_E^2$ is not fixed to 1.

such distinct extended families, each of eight individuals: two unrelated founders with two children, each with an unrelated spouse and one child of their own. The trait data are simulated according to a multivariate normal distribution with mean vector determined by an additional covariate (e.g., age), and covariance structure dictated by the relationship matrix determined by this pedigree: that is, using the model in (2), **X** is a vector of ages which are in agreement with the real data when available, or simulated at random when unavailable, and $\beta$ was set to a value of 1.5 to indicate a positive relationship between age and OST. Also, in concordance with (1), the unrelated parents have 0 covariance, each parent–offspring pair has a covariance of $0.5\sigma_A^2$; and the extended relationship pairs have covariances determined similarly.

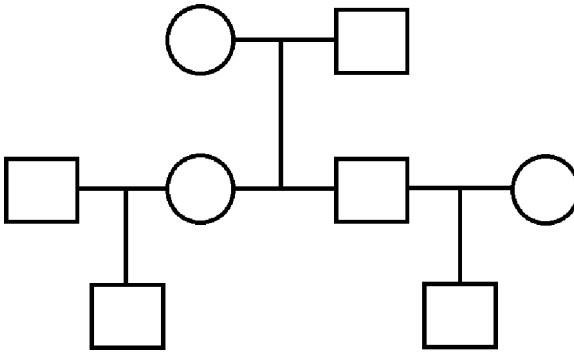

**Figure 2 Three generation pedigree.** The simpler scenario used for some simulations. Our simulated data consist of 40 repeated independent iterations of this pedigree structure, for a total sample size of 320.

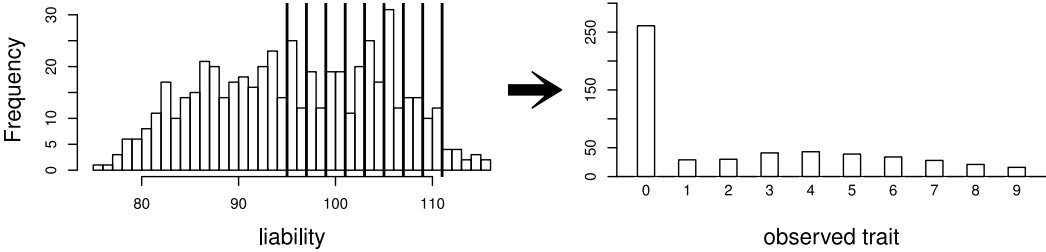

**Figure 3 Simulating a zero-inflated trait.** On the left-hand side is one simulated realization of a normally distributed liability trait, with cut-points shown for the transformation to the observed zero-inflated ordinal trait.

Using this same pedigree, we also simulate data according to the threshold model. First, a latent variable is simulated according to a multivariate normal distribution with the same mean and covariance structure as described above. This is followed by discretization of the latent variable into categories. While we explore inference with various numbers of categories, our primary interest is in a discretization into 10 categories, to mimic the actual data that we observed in the pig-tailed macaques. Specifically, the discretization is done in such a way to reflect the zero-inflated nature of our data. A graphical representation of this is shown in Fig. 3.

We also consider the pedigree of our actual data of 542 pig-tailed macaques, with multivariate normal trait data simulated with covariance structure dictated by this pedigree structure. Again, we consider simulation of both a normally distributed trait, and a zero-inflated ordinal trait dictated by a normally distributed latent variable as per the threshold model (again represented by Fig. 3). Under each scenario, four "true" heritabilities are considered: $h^2 = 0.4, 0.6, 0.75, 0.90$. The number of simulated datasets for each value of heritability is 200.

### WaNPRC pig-tailed macaques

The study population consists of six generations of pedigree data for 542 pig-tailed macaques at the University of Washington National Primate Research Center (WaNPRC).
Phenotypic data are available for 189 female monkeys present at the center in 2002, between the ages of 4.7 and 29.2 years at that time with a mean of 10.11 years old. Younger monkeys are over-represented (with $n = 45$ for monkeys between the ages of 4.7 and 6 years), and older monkeys are under-represented (with $n = 12$ for monkeys between the ages of 17 and 29.2 years).

As a proxy for DDD, we measured osteophytosis (OST), also known as bone spurs. OST trait values for each monkey were determined through radiography at each of a total possible 16 intervertebral spaces through each monkey's spinal cord, and each space was recorded as 0, 1, 2, or 3 for unaffected, slight, moderate or severe bone changes, respectively. Details of the data collection and primate facility can be found in the study by *Kramer, Newell-Morris & Simkin (2002)*.

From these raw data, there are a number of possible ways to summarize them into one number per monkey to use as the putative outcome trait. Perhaps the most obvious choice, the simple sum of the values from all intervertebral spaces, was removed from consideration because each monkey had data from a different number of the 16 total intervertebral spaces recorded; thus, there would be an upward bias in this value corresponding to the monkeys which had more spaces recorded. Therefore, we choose to focus on a subsample of the intervertebral spaces for which a large majority of the monkeys had complete data. Specifically, with the seven intervertebral spaces from location L5 to T10, there are a total of 173 of the 189 monkeys with complete data on these spaces. We then combined adjacent categories that had less than three monkeys, to give a phenotype which has a total of 10 ordered categories.

## RESULTS AND DISCUSSION

### Simulations: comparison of methods

The simulations were performed to assess both the consequences of assuming a normal distribution on an ordinal trait with normal liability, and also the performance of threshold model estimation under extreme discretization (e.g., our zero-inflated data). Under both pedigree structures, we first simulate a trait under multivariate normality with covariance structure dictated by the respective pedigree, and then perform heritability estimation of that trait under both maximum likelihood and Bayesian methods with a normality assumption. Results for the simulations under normality are shown on the left half of each panel in Fig. 4. Next, we simulate a latent trait under multivariate normality again with covariance structure dictated by the respective pedigree, and then discretize the latent trait as described earlier. We then perform heritability estimation under both maximum likelihood and Bayesian methods, but now the Bayesian method assumes the threshold model via OPR, while maximum likelihood still assumes normality. The aim of this experiment is to illustrate the potential consequences of incorrectly assuming a normal distribution, when the trait actually follows the threshold model. Results are shown on the right half of each panel in Fig. 4. Also, trace plots for chains initialized using different starting points are shown in Fig. 5 for one representative simulation scenario (WaNPRC pedigree with $h^2 = 0.60$), showing no sign of nonstationarity in each case. The starting

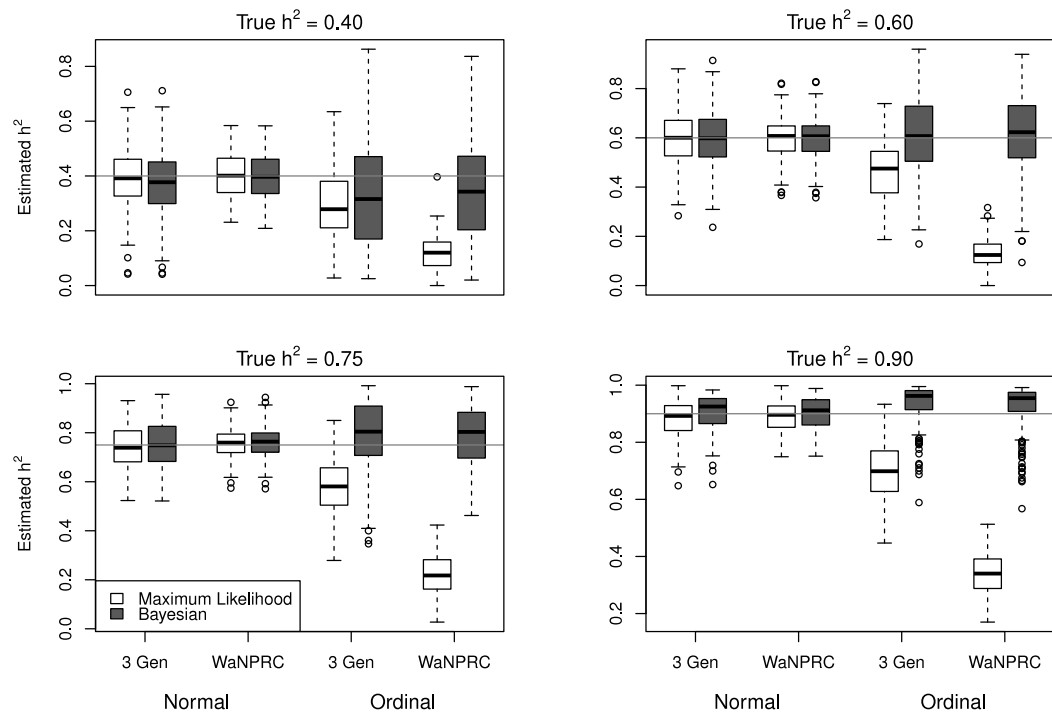

**Figure 4 Comparison between maximum likelihood and Bayesian methods.** Data were simulated both under normality (left half of each panel) and the threshold model (right half of each panel). Under normality, both maximum likelihood and Bayesian methods correctly assume normality. Under the threshold model, maximum likelihood still (incorrectly) assumes normality, whereas the Bayesian method correctly assumes the threshold model.

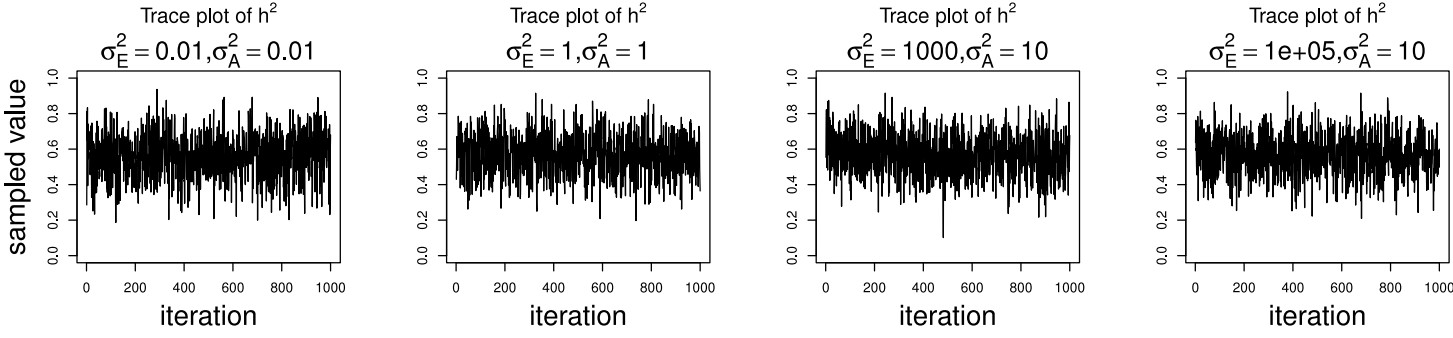

**Figure 5 Trace plots of heritability.** Chains for various starting values, for the scenario with $h^2 = 0.60$ using the WaNPRC pedigree. The values of $\sigma_E^2$ and $\sigma_A^2$ above each panel represent the starting values for the MCMC chain. Iterations were thinned at every 1000.

values for $\sigma_E^2$ varied from $(0.1, 1, 1000, 100000)$, and the starting values for $\sigma_A^2$ varied from $(0.1, 1, 10)$ as indicated on the plots. Starting values for $\beta$, **t** and **U** are obtained heuristically as described in *Hadfield (2010)*.

Under the scenarios with a normally distributed trait, maximum likelihood and Bayesian estimations both show estimates that are centered around the true values of heritability. In the scenarios with an ordinal trait, maximum likelihood gives estimates that are quite far from the true values of heritability, tending to underestimate it severely.

Table 1 **Descriptive statistics.** The mean, median, minimum, maximum, and standard deviation of each variable in the dataset are shown here.

|  | Mean | Median | Min | Max | sd |
|---|---|---|---|---|---|
| OST | 1.64 | 0 | 0 | 9 | 2.79 |
| Age (years) | 9.83 | 7.40 | 4.70 | 29.20 | 5.08 |
| Body mass (kg) | 7.06 | 6.92 | 4.53 | 12.35 | 1.40 |
| Parity (#) | 2.13 | 1 | 0 | 15 | 3.03 |

Table 2 **Heritability estimates.** Adjusted for age, mass and parity.

| Trait | Model | $\hat{\sigma}_A^2$ | $\hat{\sigma}_E^2$ | $h^2$ | 95% CI |
|---|---|---|---|---|---|
| Average OST | ML normal | 0.0394 | 0.0781 | 0.335 | $(-0.089, 0.760)$ |
| Average OST | Bayes normal | 0.0400 | 0.0815 | 0.326 | $(0.0364, 0.717)$ |
| Binary OST | Bayes OPR | $6.45 \cdot 10^8$ | $8.31 \cdot 10^8$ | 0.430 | $(1.70 \cdot 10^{-12}, 1)$ |
| Ordinal OST | Bayes OPR | $1.06 \cdot 10^{10}$ | $4.53 \cdot 10^9$ | 0.700 | $(5.56 \cdot 10^{-11}, 1)$ |

**Notes.**
For maximum likelihood, CI, Confidence Interval; for Bayesian, CI, Credible Interval.

The Bayesian OPR performs much better under these scenarios, showing estimates that are closer to the true values. This is as expected, as the OPR in fact assumes the "correct" model under these simulations. In most of the scenarios, the medians of the heritability estimates from OPR are within roughly 5% of the true value used for the simulations. We do note that under the scenario with true $h^2 = 0.90$, the estimates are centered above the true value, close to 1. Examination of some trace plots showed that the chain for $\sigma_E^2$ tended to be equal to exactly 0 for a substantial portion of the iterations, thus leading to sampled values of $h^2 = 1$ (not shown). It is thus possible that under such a high value of $h^2$, MCMC has a hard time approximating the posterior distribution of $h^2$.

## Data analysis: WaNPRC pig-tailed macaques

Descriptive statistics for the study population of pig-tailed macaques are shown in Table 1. Skewness in OST, age and parity are evident, as the mean is less than the center of the range in each case. For age, those between 5 and 6 years old are over-represented ($n = 40$), and those between 18 and 29 are under-represented ($n = 11$). For parity, 86 of the 173 monkeys had a value of 0.

The OST trait distribution is shown as the darkest bars in Fig. 6 (the left-most of each value). All analyses were adjusted for age, mass and parity, according to results from a previous study indicating that these may be potential confounders of the association between genetic factors and OST (*Kramer, Newell-Morris & Simkin, 2002*). The first two rows of Table 2 show maximum likelihood and Bayesian results from naively using the average OST value and assuming normality. The third row shows the result from using Bayesian ordered probit regresion on the ordinal phenotype described above.

Maximum likelihood and Bayesian heritability estimates under the normality assumption are comparable (0.335 and 0.326 respectively). The Bayesian OPR on the ordinal

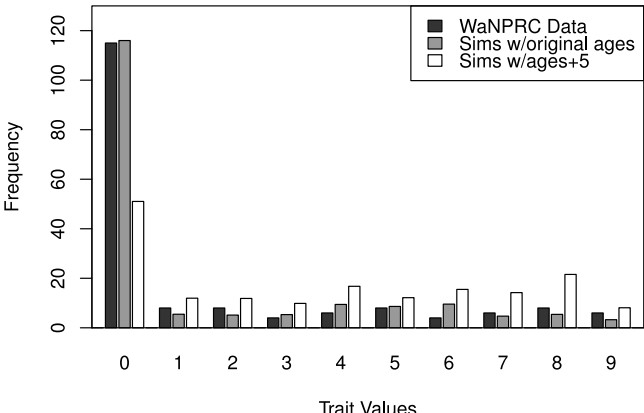

**Figure 6** Distributions of the real and simulated OST phenotype, with age shifts.

trait shows a heritability estimate that is greater (0.700), but what is remarkable is that the estimated variance components are very large ($\hat{\sigma}_A^2 = 1.06 \cdot 10^{10}$ and $\hat{\sigma}_E^2 = 4.53 \cdot 10^9$). An examination of the trace plots over MCMC generations suggested that the total variance may be unidentifiable (Fig. 1). However, this also seemed to be the case in the ordinal simulations with both the three generation pedigree and WaNPRC pedigree, where the quantity of heritability was recovered successfully (as estimates tended to be centered near the true values, as shown in Fig. 4). While this is of some technical concern, it thus seems more important for our current purposes to examine the posterior distribution of heritability as estimated from the MCMC. In our real data, we find that the posterior distribution simply reflects the information provided by the prior; that is, our estimation procedure was not able to extract substantial information from the data. This is shown in panels A and A.1 in Fig. 7. A similar posterior distribution of heritability was observed in the binary case (not shown). Also, results using different prior distributions are shown in subsequent rows of Fig. 7. We observe that with $n = 173$ in either the real data or simulated case, the estimated posterior distributions tend to reflect the prior distributions. In some cases, mixing appears to be good, in the sense that the MCMC chain travels between 0 and 1 with no discernible pattern, such as with the Beta(0.01, 0.01) or Beta(0.1, 0.1) priors using the WaNPRC data. In other cases, mixing appears to be poor, such as with a Beta(0.2, 0.2) prior using the WaNPRC data, or the Beta(0.01, 0.01) prior using the simulated dataset with $n = 173$; in these cases, the posterior distribution reflects one of the two modes of the prior distribution. These cases do raise uncertainty as to whether the posterior distribution is simply hard to estimate here, or if the posterior distribution truly contains no information about $h^2$. However, with increased sample size such in the three panels with $n = 542$, we obtain much stronger indications of stationarity of the MCMC chain in all cases, and unimodal posterior distributions of $h^2$, thus leading us to hypothesize that the true posterior distribution of $h^2$ contains more information about $h^2$ with larger sample sizes. The dataset and R code are provided in the Supplemental Information.

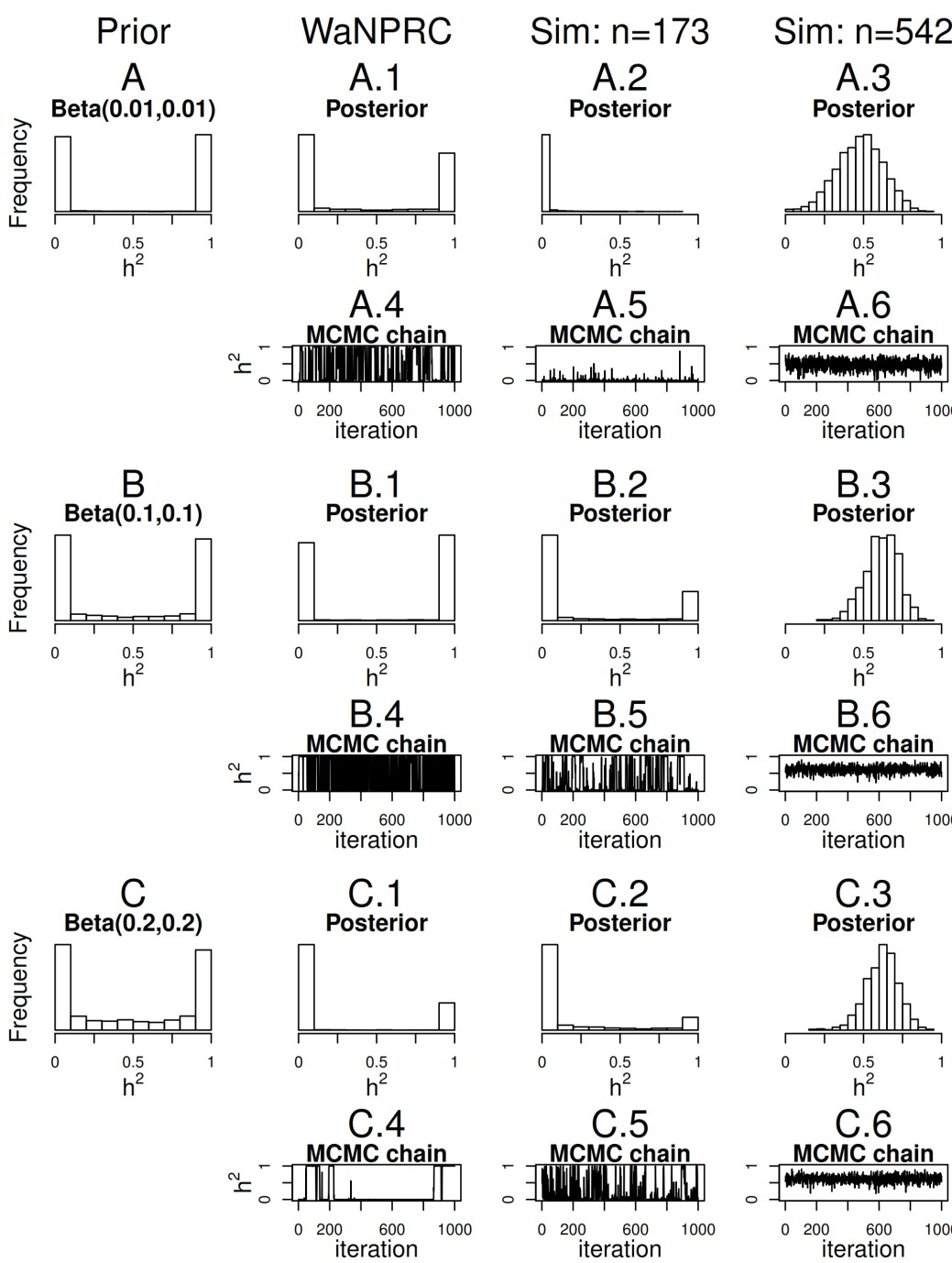

**Figure 7 Distributions of heritability.** Three scenarios with different prior distributions are shown consecutively, with two rows of panels for each scenario. (A–C) show empirical realizations within each scenario of the prior distributions of heritability, according to inverse-gamma prior distributions on each of the individual variance components. (A.1, B.1, C.1) show the posterior distributions of heritability from the real data analysis. (A.2, B.2, C.2) show the posterior distributions of heritability from 173 simulated monkeys, and A.3, B.3 and C.3 show the posterior distributions of heritability from 542 simulated monkeys. A.4–A.6, B.4–B.6, and C.4–C.6 show trace plots of heritability corresponding to each scenario, thinned to 1000. Simulated heritability was 0.60 in each case.

## Sample size exploration

Since we were not able to extract any conclusive information from our data, we explored simulations to determine how much data would be necessary for heritability estimation under the threshold model. First, we simulate two extreme cases: 173 monkeys (identical to that of our real data), and the full 542 monkeys in the entire WaNPRC pedigree. In each case, a zero-inflated trait is simulated under the threshold model. Again, we focus on the posterior distributions of heritability, which are shown in A.2–A.3, B.2–B.3 and C.2–C.3 of Fig. 7.

As shown, with 173 monkeys, threshold model heritability estimation typically produces estimated posterior distributions that mimic the prior distribution, even when the data are simulated according to the same threshold model that we are using for estimation. In contrast, with the full pedigree of 542 monkeys, threshold model heritability estimation succeeds in producing a spread of MCMC samples around our truth of $h^2 = 0.60$. We next aimed to determine the minimal number of monkeys required to estimate heritability under simulation.

Our criteria for labeling a particular sample size as having estimable heritability follows from our observations with the sets of 173 and 542 monkeys. That is, we examined the resulting estimated posterior distributions of $h^2$ at each sample size. Specifically, we checked the proportions of the estimated posterior distribution that fell into each of the 10 bins of size 0.1, from 0 to 1. Then, if the bins of 0–0.1 and 0.9–1.0 had the smallest proportion of mass from the estimated posterior distributions, we determined that the sample size had estimable heritability. In each such case, we also observed a unimodal posterior distribution with its mode near the simulated true $h^2$, so while our criteria only depends strictly on the decreasing tails of the posterior distribution, the result is that a sensible posterior distribution of $h^2$ indicates that $h^2$ is estimable.

To this end, we created subsets of the full WaNPRC pedigree, proceeding by starting with the original 173 monkeys and adding the most related monkeys to that set, based on cumulative pairwise kinship coefficient. That is, the candidate monkey who is the "most related" to the current set would be the one who has the greatest sum of kinship coefficients with each monkey in the set, and is not currently in the set itself. Also, 28 of the 173 monkeys actually are not related to any of the others in this set, so these were first removed. We then added monkeys based on the maximum kinship criteria to create larger subsets of monkeys (e.g., $n = 200$, $n = 210$, etc.), and proceed with our simulations as if these were the monkeys for which we had data. We note that with sample sizes for which $h^2$ appeared to be estimable, stationarity of the MCMC was typically observed within roughly 1 million iterations, at which point the above criteria for estimable heritability was always satisfied. For sample sizes in which the posterior distribution did not satisfy our criteria for estimable heritability, the trace plot for $h^2$ would typically appear similar to the prior distribution of $h^2$, with trace plots showing no sign of nonstationary behavior by the MCMC chain, as it bounces back and forth between 0 and 1 (such as in select panels of Fig. 7). Additionally, when we ran certain scenarios with insufficient sample sizes for up to 200 million iterations, the trace plots for $h^2$ appeared the same as at 1

million iterations, adding further evidence that the chain's repeated jumping from 0 to 1 exhibits its stationary behavior. This suggests that our MCMC appears to be providing a good approximation of the true posterior in both cases when the sample sizes lead to estimable $h^2$, and when sample sizes are low, with the true posterior not containing much information about $h^2$.

Additionally, we wanted to explore the effect of attenuation on the degree of zero-inflatedness in our trait distribution, and whether a less extreme distribution may lend itself to better heritability estimation. This has direct relevance to our real OST phenotype, as it is a trait which manifests itself gradually over the lifespan of monkeys: in an older sample of monkeys we expect to see a less zero-inflated trait distribution. By simply increaing the value of our age covariate in our simulations by five years for each monkey, we obtain this effect. An illustration of the trait distribution resulting from the five-year age increase is shown in Fig. 6, with empirical averages from 100 datasets for the simulated cases. Based on the posterior distribution histograms of heritability (not shown, but similar to A.2–A.3, B.2–B.3 and C.2–C.3 of Fig. 7), we determine whether there was enough information in the simulated data for each case.

Alternatively, we also examine phenotype data from another population of monkeys, in the Wisconsin National Primate Research Center. These monkeys are older than our WaNPRC center monkeys, with a mean age of 21.55 years old. Therefore, almost all of the monkeys have exhibited some degree of the OST trait and there is no zero-inflatedness. We perform simulations with a trait distribution that mimics this, to again determine what sample size is required for estimable heritability.

Although it has less relevance to our primary WaNPRC data, we also explore whether having phenotype data on a different subset of monkeys than the original 173 may be more optimal, with regards to heritability estimation. That is, thus far we have merely added additional monkeys to the original set in which our real dataset has phenotype data. However, these original monkeys are not all highly related to each other, which provides less information for heritability estimation than if they were all highly related. Thus, it is also of interest to know whether a smaller sample size would be necessary to obtain estimable heritability under a more related set of monkeys. We therefore sample monkeys based on maximum cumulative kinship coefficient starting from the single monkey which is the most related to all other monkeys under this criteria, and add monkeys as before. We simulate data under both the original trait distribution, and that of the Wisconsin dataset.

Finally, we explore sample size requirements under the more simple three-generation pedigree, i.e., the same one as in our previous simulations shown in Fig. 4. These previous simulations were performed with a sample size of 40 families, or 320 individuals. We find that we can reduce the sample size to 20 families, or 160 individuals, and still obtain reasonable heritability estimation through the threshold model. Also, with a trait distribution that is less zero-inflated (again as through an increased age by 5 years), not much improvement is obtained; we can further reduce the sample size by just one family, to 19 families or 152 individuals. A summary is shown in Table 3, and R code for one simulation scenario is provided in the Supplemental Information.

**Table 3  Minimum sample size required for estimability of heritability under the threshold model.**

| Pedigree | Phenotyped | Trait distribution | Min. sample size |
|---|---|---|---|
| WaNPRC | Original | Original | 250 |
| WaNPRC | Original | Age + 5 years | 230 |
| WaNPRC | Original | Wisconsin | 250 |
| WaNPRC | Optimal | Original | 190 |
| WaNPRC | Optimal | Wisconsin | 180 |
| Three generation | All | Original | 160 |
| Three generation | All | Age + 5 years | 152 |

## Discussion

Here, we examine heritability estimation of an ordinal trait. Our ultimate aim is to determine whether osteoarthritis is heritable, and we explored a number of modeling considerations that take account of the ordinal nature of the data that were collected. We discovered that heritability estimates can vary greatly based on the choice of model, from both our simulation study and our real data analysis. In our WaNPRC macaques, under the naive assumption of normality of the average OST value, we observed an estimate that indicates a slight-to-moderate amount of heritability (0.335 under maximum likelihood estimation). This is also observed in the Bayesian estimate, under the same model (and with non-informative priors).

However, our simulations illustrate the degree to which inference can be biased, if normality is assumed when the data actually follow the threshold model. Ordered Probit Regression was able to obtain heritability estimates that were centered closer to the true value in each case than maximum likelihood estimates under the normality assumption. While it is no surprise that Ordered Probit Regression was able to obtain good estimates from these datasets since they were simulated under the exact model that the Ordered Probit Regression assumes, it is more to the point that using a standard maximum likelihood approach with an assumption of normality yielded estimates that were quite far from the true values, even when the number of categories was large (e.g., 10 in two of the scenarios).

These scenarios were also designed to mimic a plausible imitation of our real data, in the fact that most of the monkeys (115 out of 173) were "normal" with respect to the second OST trait. We simulated the liability trait and then put the bulk of the data into the first category in order to attain a similar distribution of the observed trait. Whether or not this model exactly reflects the biological mechanism of the OST trait, these simulations nevertheless illustrate that incorrectly assuming normality of an ordinal trait invites the risk of producing misleading heritability estimates, while Ordered Probit Regression has a better chance of producing estimates that are closer to the truth. Furthermore, while it is true that we do not know whether our actual data follow the threshold model, this assertion could be justified by applying the polygenic model to the liability; that is, even if what we observe is ordinal with a very non-normal distribution, it is defensible to assume

that, if the trait is determined by many loci, there may be an underlying latent variable which does have an approximately normal distribution.

Thus, it is interesting that our heritability estimate rises to 0.700 under estimation with the threshold model. However, there are several alarming aspects to this: (1) the estimates of the individual variance components are very large; (2) the 95% Credible Interval spans essentially the entire range of (0, 1); (3) the posterior samples of heritability almost exactly mimic its prior distribution. These observations suggest that the information content of our data is not high, which may be surprising given that we do have 173 monkeys with trait data. However, as our regression setting here is a non-standard one, we find it useful to perform simulations to explore how much is required to obtain estimable heritability.

For the sake of its direct relevance to our real data, we first examine the effect of increasing the sample size on our actual WaNPRC pedigree. Our original sample of phenotyped monkeys was a convenience sample that was not specifically intended for heritability estimation, and many of the monkeys which were not originally phenotyped are still alive and could still be obtained. Obtaining these data from another 80+ monkeys, however, is non-trivial, and we are still investigating this possibility.

It is somewhat surprising that we do not gain much improvement in sample size reduction with a more balanced trait distribution that was induced by shifting the age distribution. It is possible that there are nuances in our simulated trait distributions which are causing difficulties that we do not understand, particularly because all of our threshold locations were placed in an *ad hoc* manner, simply to create trait distributions that appeared reasonable. On the other hand, it is also possible that the threshold model does not inherently struggle with zero-inflated data (at least when such data truly arose from the threshold model itself), and so an improvement is not to be expected with less zero-inflated data. This possibility is corroborated by the fact that, using the Wisconsin trait distribution, we also see limited and/or no improvement to sample size requirements, depending on the set which was phenotyped. It is thus interesting to note that the actual trait distribution seems to be far less of a factor than the set of monkeys for which phenotype data are available, in terms of obtaining estimable heritability.

Of the previously mentioned alarming aspects to our heritability estimate on our real data, the one that our simulations does not address is that of the extremely high estimates of the individual variance components. In fact, this seems to be a recurring observation even when the posterior distribution of heritability appears to be well-behaved. While we are reasonably satisfied to simply obtain sensible posterior distributions of heritability from this implementation of the threshold model, this suggests that the individual variance components in fact are not identifiable in our scenarios. A resolution to this concern is a possibility for further study.

## CONCLUSIONS

Although it is no surprise that model misspecification can result in biased estimates, the extent to which this may be true with heritability estimation of an ordinal trait has not been demonstrated in the literature, to our knowledge. Thus, we illustrate the severe biases

that may result when a normality assumption is made on data which follow the threshold model.

Next, we perform a real data analysis to estimate heritability of osteoarthritis in pig-tailed macaques. Unfortunately, we determine that our dataset does not have enough monkeys in order to obtain reliable estimates of heritability, despite a seemingly adequate sample size of $n = 173$.

Thus, we conclude with an examination of sample size requirements in this setting, via simulation. We do this under a variety of scenarios, using both our real data WaNPRC pedigree and also a simpler pedigree structure with three generations and eight individuals (Fig. 1). Under the WaNPRC pedigree, we find that somewhere between roughly 180 and 275 monkeys are required to obtain estimable heritability, depending on the trait distribution and relatedness of the phenotyped monkeys. Under the three generation pedigree, we find that roughly 160 monkeys (20 independent families) are required to obtain estimable heritability. These results should prove to be useful to biologists and other researchers who are planning to study the heritability of an ordinal trait.

## ACKNOWLEDGEMENTS

We thank Peter Hoff for useful discussions regarding identifiability in probit models.

### Funding

VNM was supported by the National Scientific Foundation grant DMS 0856099 and National Institutes of Health grant IRC4AI092828-01. PAK was supported by the National Institutes of Health grant U01 AG21379. The funders had no role in study design, data collection and analysis, decision to publish, or preparation of the manuscript.

### Grant Disclosures

The following grant information was disclosed by the authors:
National Scientific Foundation grant: DMS 0856099.
National Institutes of Health grant: IRC4AI092828-01.
National Institutes of Health grant: U01 AG21379.

### Competing Interests

The authors declare there are no competing interests.

### Author Contributions

- Peter B. Chi analyzed the data, contributed reagents/materials/analysis tools, wrote the paper, prepared figures and/or tables, reviewed drafts of the paper.
- Andrea E. Duncan conceived and designed the experiments, performed the experiments, analyzed the data, contributed reagents/materials/analysis tools, wrote the paper, reviewed drafts of the paper.

- Patricia A. Kramer conceived and designed the experiments, performed the experiments, contributed reagents/materials/analysis tools, wrote the paper, reviewed drafts of the paper.
- Vladimir N. Minin contributed reagents/materials/analysis tools, wrote the paper, prepared figures and/or tables, reviewed drafts of the paper.

### Supplemental Information

Supplemental information for this article can be found online at http://dx.doi.org/10.7717/peerj.373.

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
