# Peer review of "Heritability estimation of osteoarthritis in the pig-tailed macaque (Macaca nemestrina) with a look toward future data collection"

_PeerJ, doi:10.7717/peerj.373_

## Round 0.1 · original submission · Minor Revisions

Dear authors, please revise the manuscript according to the comments.

Reviewer 1 ·

Basic reporting

Missing one equation number on page 4, the first equation (line 17) on this page is missing number

Experimental design

The paper is well constructed and the points are clearly presented.
There is one concern about the design. The author mentioned that the age is not very well normally distributed in the real data(page 6&7, over presented for young population and under presented in old population). While in general population, age is not normally distributed, it is skewed. So in such case, I wonder if the author should apply some transformation or state the skew is not severe enough to affect the results

Validity of the findings

The dataset are clearly stated in the paper. The author provided a proof study on a well-known point, but no published paper talking about this. One suggestion about the results show in figure 1, the convergence might be presented with a p-value.

Additional comments

Overall this is a well presented paper. All points are clear. Data sets used in the paper are clearly stated. The reviewer just has some concern about whether the distribution of covariates will affect the results or not. The convergence test p-value for the data shown in figure 1 might help understand the figure better. And only a missing index of the equation is found on page 4.

Reviewer 2 ·

Basic reporting

This manuscript was well written. The study on heritability estimation of osteoarthritis is thorough and impressive, even though no confirmative conclusions were obtained on the real data, due to the limitations on the sample size and relatedness. The authors went further to explore the minimum sample sized required for estimation of heritability. Their findings in this work will be very useful for future studies. Though no additional experiments are needed for this study, I am excited to see how their models will perform when they have enough samples in their future work.
A few minor issues:
1. No sequential line numbering makes annotation a little bit difficult.
2. Spell out the entire phrases the first time you use them. On page 2, MCMC in “e.g. improving the MCMC convergence…”; On page 5, OST in “and β was set to… and OST”.
3. Use a, b, c… to indicate the individual panels in the multi-panels figures, and change figure captions and the main text accordingly.
4. Axis labels in Figure 1, 5 and 7 are too small to read.

Experimental design

The experiments were well designed and the methods were described in detail.

Validity of the findings

The conclusions looks reasonable.

Additional comments

No comments.

Reviewer 3 ·

Basic reporting

This manuscript is well written and structured and conforms the PeerJ template.

Experimental design

The research questions in this manuscript is clearly stated. The data and method sections are described with sufficient and reproducible details in terms of equation, explanation and software.

Validity of the findings

The authors has made sufficient efforts to investigate their data. Although the conclusion is inconclusive, which is possibly due to the sample size, the results still have potential impact.

Additional comments

The major point of this manuscript is to investigate scenarios in which complex trait data violate normality assumption during heritability estimation. Using both simulation and real data, authors compared these two models and were reluctant to make decisive conclusion due to insufficient data. Some concerns about this study are:

(1). To what degree, the selection of cut-points during data transformation will impact the heritability estimation. In Figure 3, how did authors select cut-points? Have they ever tried other cut-point thresholds in later stage of heritability estimation?

(2). In figure 4, can authors provide any p-values to test the deviation of OPR estimation to true values? Can authors also propose any solution to heritability estimation in context of high real heritability values in the "Simulations: Comparison of methods" section.

---

## Round 0.2 · accepted · Accept

Dear authors,

After review your revisions, we decide to accept your manuscript for publication. Thanks for submitting your manuscript to PeerJ. We are looking forward for more submissions from you.